



# Assessing road segment impact on accessibility to critical services in case of a hazard

Sophie Mossoux[1,2], Matthieu Kervyn[1], Frank Canters[2]

[1]Physical Geography, Department of Geography, Earth System Science, Vrije Universiteit Brussel, Pleinlaan 2, 1050 Brussels, Belgium
[2]Cartography and GIS Research Group, Department of Geography, Vrije Universiteit Brussel, Pleinlaan 2, 1050 Brussels, Belgium

*Correspondence to*: Frank Canters (frank.canters@vub.be)

**Abstract.** Development of hazard maps is one of the measures promoted by the international community to reduce risk. Hazard maps provide information about the probability of given areas to be affected by one or several hazards. As such they are useful tools to evaluate risk and support the development of safe policies. So far studies combining hazard mapping with accessibility to services are few. In hazardous environments, accessibility of the population to strategic infrastructure is important because emergency services and goods will principally be offered at or provided from these locations. If a road segment is blocked by a hazard, accessibility to services may be affected, or worse, people may be completely disconnected from specific services. The importance of each road segment in the transport network as a connecting element enabling access to relevant services is therefore critical information for the authorities. In this study, we propose a new application of hazard mapping which aims to define the importance of each road segment in the accessibility to services, taking in account the probability of being affected by a hazard. By iteratively removing one segment after the other from the road network, changes in accessibility to critical infrastructure are evaluated. Two metrics of road segment importance considering the population affected and the hazard probability are calculated for each segment: a road accessibility risk metric and a users' path vulnerability metric. Visualization of these road metrics is a useful way of valuing hazard maps and may help to support discussions about the development of new infrastructure, road capacity increase and maintenance of existing infrastructures, and evacuation procedures.

## 1. Introduction

A well-developed transport network is essential to the smooth running of a country since it plays an important role in supporting social and economic activities (Hong et al., 2015; Jenelius et al., 2006; Mattsson and Jenelius, 2015; Nagurney and Qiang, 2012). A reliable road network is even more important in a hazardous environment where the connection of villages to strategic infrastructures such as hospitals, fire stations, and commercial and employment centers must be guaranteed, particularly when a hazard occurs. Even if roads are considered as essential infrastructure, the road network is often designed to function close to maximum capacity to minimize the costs, with small margins of reserve capacity and little redundancy (Mattsson and



Jenelius, 2015; Taylor et al., 2006). It is therefore sensitive to potential disruptions and its interdependencies with other systems can lead to serious consequences for the functioning of society and economic activities (Hong et al., 2015; Mattsson and Jenelius, 2015).

Given the functional value of the road network, studies have been conducted to assess road network related vulnerability to a

disruption from different points of view. Some metrics have been proposed to characterize vulnerability of the population to road disruptions at different administrative levels (e.g. municipalities, states) (Jenelius et al., 2006) and accessibility to main road axes before and after the occurrence of a natural hazard (Sohn, 2006; Taylor and Susilawati, 2012). Other studies assess the robustness of the road network as a whole, defined as the degree to which the system can function correctly according to its design specifications in the presence of serious disruptions (Bil et al., 2014; Chang and Nojima, 2001; Immers et al., 2004;

Institute of Electrical and Electronics Engineers, 1990; Nagurney and Qiang, 2012; Sullivan et al., 2010). At a more local scale, previous research has characterized vulnerability at the level of individual road segments. In these studies potential degradation of the road transport system caused by interruption of a specific road segment and its impacts on society is analyzed (Jenelius and Mattsson, 2015). Through several metrics, studies have evaluated the direct physical, economic and functional impacts of a road segment disruption (Blake et al., 2017; Pregnolato et al., 2017; Winter et al., 2016) as well as indirect impacts,

by analyzing how users adapt their way of travelling in case of a disruption (Bil et al., 2014; Jenelius and Mattsson, 2015; Postance et al., 2017; Taylor et al., 2006).

Several types of disruption can lead to a road failure (e.g. accidents, technical failures, hazards or antagonistic actions). Natural hazards, events such as floods (e.g. Hong et al., 2015; Sohn, 2006), landslides (e.g. Postance et al., 2017), earthquakes (e.g. Chang and Nojima, 2001; Peeta et al., 2010), ash fall (e.g. Blake et al., 2017) or lava flows, can cause serious perturbation as

roads may be interrupted or road conditions may deteriorate (Mattsson and Jenelius, 2015). Road disruptions are often integrated in hazard studies through scenarios which focus on specific locations of the network. In such approach, hazards may be modelled within administrative entities based on the selection of one possible scenario (Hong et al., 2015; Mattsson and Jenelius, 2015). Other studies define the road segments that are most susceptible to be affected depending on their location relative to historical hazard zones (Sohn, 2006) or their closeness to areas having a high susceptibility to host a hazard (Postance

et al., 2017). Probabilistic hazard maps provide relevant information about the probability of given areas to be affected by a hazard and methods to produce such maps have developed significantly over the last years (e.g. for lahars (Bartolini et al., 2014; Sandri et al., 2014), landslides (Alexakis et al., 2014), earthquakes (Yazdani and Kowsari, 2017), pyroclastic flows (Bartolini et al., 2014; Sandri et al., 2014; Tierz et al., 2016), lava flows (Becerril et al., 2014; Favalli et al., 2009, 2012), tephra (Becerril et al., 2014; Bonadonna et al., 2005; Sandri et al., 2014)…). Such maps are produced by combining data on historical

events with physical or statistical modelling (Calder et al., 2015). Surprisingly, no study is known to us that integrates information provided by probabilistic hazard maps in the assessment of impacts over an entire transport network.

To address this gap, this study proposes two metrics to characterize the importance of each segment in a road network in terms of accessibility to services in case a natural hazard occurs, using spatially explicit information on hazard probability. The first metric, the "road accessibility risk", combines the road segment's potential impact on the accessibility and travelling time of



the population to the closest infrastructure ("road access vulnerability") with the probability of occurrence of the hazard ("road hazard exposure"). The second metric, the "users' path vulnerability", considers, instead of the travelling time, the reliability of the alternative path the user needs to follow in case a road segment becomes disrupted or, in other words, the confidence that a user is not being impacted by the hazard when attempting to access the closest facility. We illustrate the use of the

proposed metrics on Ngazidja Island (Union of the Comoros), a volcanic island where the road network is potentially exposed to lava flows. Access of the population to the closest hospital is used as an example of a key infrastructure that should be within reach, both in normal conditions as well as at the time of a hazardous event.

## 2.   Study area

Located northwest of Madagascar, in the Mozambique Channel, Ngazidja is the most western island of the Comoros

archipelago (Figure 1). It is exposed to a range of volcanic hazards, including lava flows. The central part of the island is formed by the active Karthala volcano (2361 m a.s.l.) which erupted every six to eight years on average over the last 200 years (Bachèlery et al., 2016). The roads on Ngazidja Island are classified into three categories (Figure 1): national, regional and local roads (PADDST, 2014). Historical lava flows (Bachèlery and Coudray, 1993) suggest that future volcanic impacts on the road network might be severe as the majority of lava flows in the past reached the coast line. The 1858 flow, for example,

was issued at an altitude of 2100 m a.s.l. and travelled over a distance of 13 km to reach the ocean (Figure 1 ; Bachèlery and Coudray, 1993). During the 1977 eruption, an 'a'ā lava flow issued from a fissure at 360 m a.s.l. on the southwest flank, crossed the villages of Singani and Hesta, and destroyed 566 meters of the road network (Krafft, 1982).



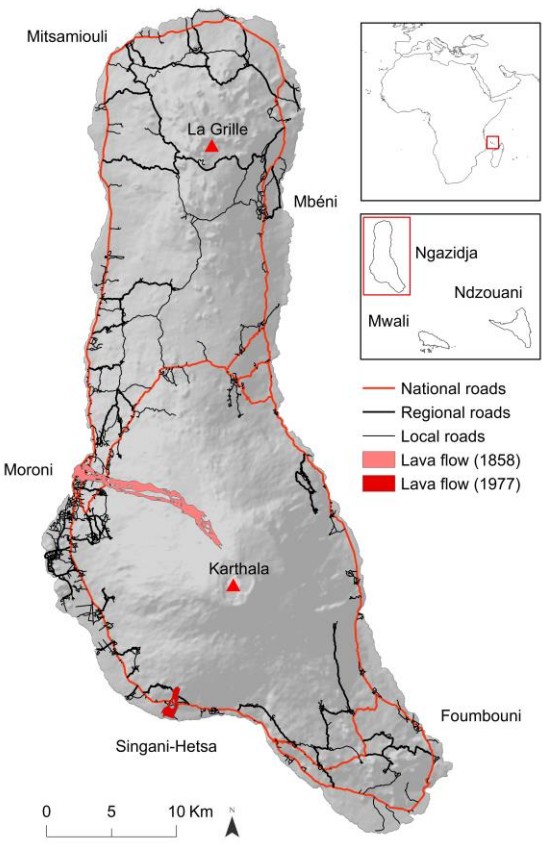

**Figure 1 - Road network (own processing) on Ngazidja Island and some historical lava flows. The insets show the location of the island in Africa and within the Comorian archipelago.**

## 3. Material and methods

5  To assess the importance of road segments in people's mobility to strategic infrastructure (i.e. hospitals) taking in account the probability of a road segment being affected by a hazard (i.e. lava flow), this research encompasses the following steps: (1) collecting data on population, strategic infrastructure and hazard probability, (2) building a digital representation of the road network and assessing each road segment's attributes, (3) evaluating the population's accessibility to the closest infrastructure under normal and disrupted conditions and (4) calculating road accessibility risk and users' path vulnerability.



### 3.1. Data collection

#### 3.1.1. Population data

It is estimated that close to 302 000 people live on Ngazidja Island (estimate for 2013; Mossoux et al., 2018). The population estimated for each village is here used. While most of the population is concentrated in villages along the coastal area, 40% of the population lives in the capital Moroni and its surroundings (Figure 2).

#### 3.1.2. Strategic infrastructure

As key infrastructure, we consider six hospitals of the island providing specialized medical services (e.g. surgery, urgency, radiology…) (Centre d'Analyse et de Traitement de l'Information, 2016). Three of the six hospitals are located in the capital and its direct surroundings whereas the other three are located in secondary cities of the island: Mitsamouili, Mbéni and Foumbouni (Figure 2). Due to a lack of data, the capacity and offered services of the hospitals are not considered in the analysis.

#### 3.1.3. Lava flow hazard map

A lava flow invasion hazard map identifies the locations that may be affected by a lava flow within a given time period (De La Cruz-Reyna et al., 2000; Sigurdsson et al., 2015; Thompson et al., 2017). The one used in this study provides for each location (corresponding with a cell of 90 m resolution) a probability of being inundated by lava flow during the next eruption (Figure 3). Probabilities have been computed using (1) QVAST, a plugin to produce a vent opening susceptibility map using information on the presence of volcanological structures (e.g. vents, fissures…) located on the volcano (Bartolini et al., 2013) and (2) Q-LavHA, an open-source plugin that simulates lava flow inundation probability from eruptive zones (e.g. vents, fissures and surfaces) on a Digital Elevation Model (DEM). Q-LavHA combines different models that determine the spatial propagation of a channelized 'a'ā lava and its terminal length using an iterative approach (Mossoux et al., 2016). It provides for each pixel a value between 0 and 1 which represents the pixel probability to be inundated during a next eruption. To produce lava flow hazard maps, Q-LavHA enables the user to consider vent opening susceptibility as calculated by QVAST and simulates lava flows from regularly distributed vents based on user defined input parameters that characterize the flow (Table 1).

**Table 1 – Lava flow hazard map input parameters implemented in Q-LavHA.**

|  | Parameters | Value |
|---|---|---|
| Input file | Digital elevation model (m) | 90 |
| Lava flow propagation | $H_c$ (m) | 3 |
|  | $H_p$ (m) | 9 |
| Lava flow length constraints | Mean length (m) | 5200 |
|  | Standard deviation (m) | 3200 |
| Vent opening susceptibility map | Distance between the vents (m) | 90 |
|  | Minimum probability to simulate | 0 |
| Simulation parameters | Number of iterations | 500 |





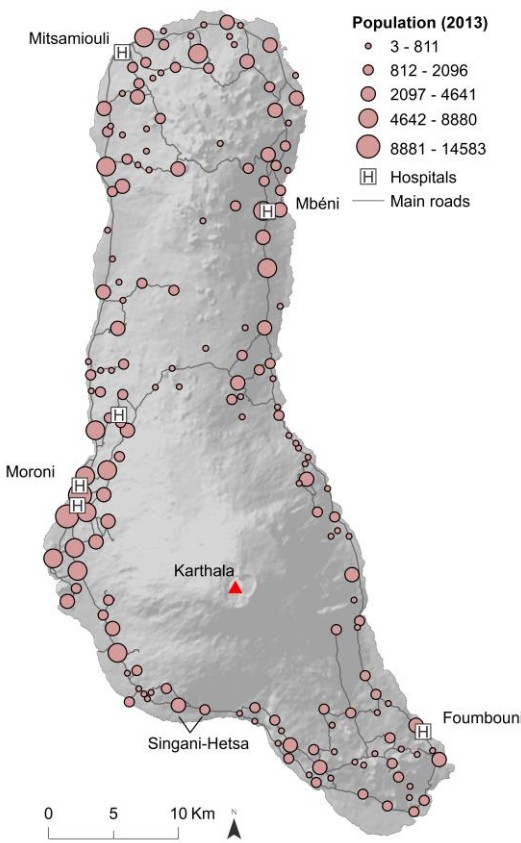

**Figure 2 - Population repartition estimated for 2013 (Mossoux et al., 2018) and location of the hospitals (Centre d'Analyse et de Traitement de l'Information, 2016) considered in the road accessibility risk and users' path vulnerability assessment.**

### 3.2. Building the road network and attributes

Because of the absence of detailed digital road network data, very high spatial resolution satellite images (Pléiades, 2013; 0.5 m spatial resolution) and GPS tracks (Garmin eTrex 30x) acquired during three field missions (summer 2013, 2014 and 2016) have been used to delineate the road network (Figure 1). Pathways are not included in the network.

The digitized road segments were converted to a network formed by nodes and links. In this network nodes represent road intersections or the location of villages and infrastructure from which accessibility will be assessed. Each village is characterized by its number of inhabitants. Links in the network correspond to road segments and are described by (1) travel time to cross the segment and (2) cumulative susceptibility of the segment of being affected by lava flow.



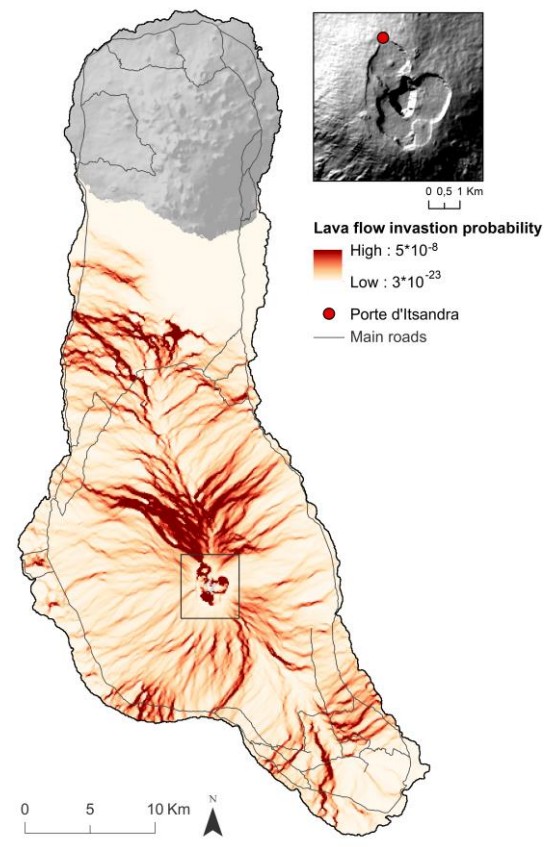

**Figure 3 - Lava flow hazard probability map of Karthala volcano (own processing). The inset provides a closer view on the caldera and the "Porte d'Itsandra" which is an opening in the North of the caldera where lava can escape.**

To calculate travel time information on travel speed is required. Because no data on effective driving speed is available for the considered road network, the GPS tracks acquired during the three field missions were used to estimate the average speed reached on each road segment (km/h). A total of 139 GPS tracks were collected covering 93% of the national roads, 62% of the regional roads and 7% of the local roads. Some road segments are covered by several GPS tracks. As each individual road segment is assigned the average speed recorded by the overlaying tracks, a spatial variation within road segments of the same

type can be observed. For segments not covered by GPS tracks, travel speed was estimated based on the average speed measured on road segments of the same type.

While the average speed reached on national road segments (speed$_{national\ roads}$ = 27 ± 10 km/h) is higher compared to other road types, it is not much higher than on regional roads (speed$_{regional\ roads}$ = 22 ± 11 km/h). Low speeds observed on national and regional roads are mostly related to the limited width and/or poor condition of the roads. The average speed on local road

segments is limited to 13 ± 11 km/h due to the sinuosity and the location of these segments within cities and villages.





Measurements are of course dependent on driver characteristics, mode of transport (e.g. personal car, taxi or public bus), time of the day and moment of the year, as well as the state of the road at the time of the acquisition. No differentiation between different modes of transport was possible. Despite these biases, the GPS measurements give a realistic impression of the local transport characteristics and are representative of the spatial variability of effective transport velocity on the road network.

These data (in km/h) were used to estimate the travel time (in seconds) over each road segment $i$ (Equation 1):

$$\text{Travel time}_i = \frac{\text{Length}_i}{\text{Speed}_i * \left(\frac{1000}{3600}\right)} .$$ (1)

Each road segment was finally characterized by its susceptibility of being inundated, and therefore blocked, by a lava flow. To calculate this susceptibility, the probabilities of the lava flow hazard map cells (Figure 3) underlying all road segments were normalized to sum to one. Then for each road segment $i$ these normalized values were summed up to produce a relative

measure of the susceptibility of the segment to be affected by a lava flow ($h_i$). A sum of the normalised probabilities is here favored, instead of calculating an average of the original probability values for each segment because this way the length of the segment is accounted for. Indeed, all other things being equal, the longer the road segment, the higher the chance will be that it could be blocked due to a lava flow.

### 3.3. Accessibility before and after disruption

Considering an undisturbed road network, each village weighted by its number of inhabitants (Mossoux et al., 2018) was, in first instance, assigned to the closest infrastructure based on Dijkstra's shortest path algorithm (Dijkstra, 1959) (Figure 4 - normal situation). The shortest path was calculated according to travel time (s).

In a second stage, each road segment was iteratively removed from the network to simulate the effect of a road segment being obstructed (Appert and Chapelon, 2007). Based on the network with one segment removed, Dijkstra's shortest path algorithm

was applied again to define the new shortest path to the closest infrastructure from each village (Figure 4 - disrupted situation).

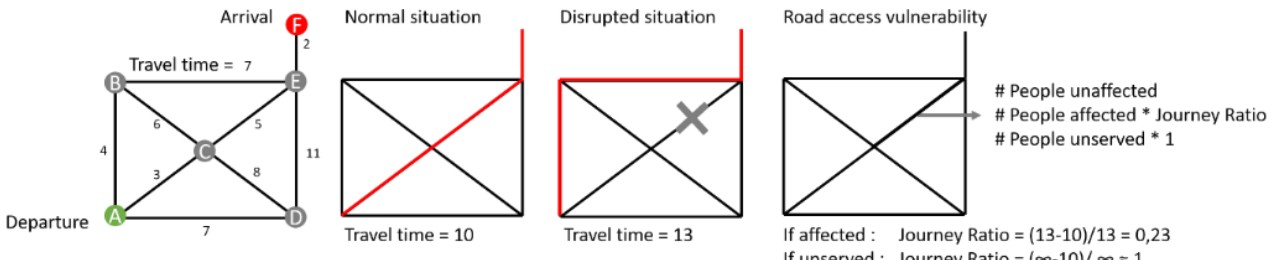

**Figure 4 - Accessibility assessment before and after a disruption using Dijkstra's shortest path algorithm. People affected and unserved are weighted by the Journey Ratio which assesses the travel time increase caused by the disruption and re-routing.**

Comparison with the undisturbed road network allows defining the overall impact of the road segment closure on the accessibility of the population to the closest infrastructure. Three scenarios are possible:





- Inhabitants of a village are unaffected by the road removal. The closest infrastructure and the travel time are still the same as in the undisturbed situation.
- Inhabitants of a village are affected ($pop_{affected}$). The road removal increases the journey in time or assigns the road users to a new infrastructure which is located further away.
- Inhabitants of a village become isolated from available services and remain unserved ($pop_{unserved}$). They no longer have access to any of the present service infrastructures (Jenelius et al., 2006).

At the end of the process, each road segment is characterized by the total number of people it would affect or isolate if the road segment would become obstructed (Figure 4). Additionally, the travel time (s) and the hazard exposure experienced by the affected population before and after the road segment obstruction is recorded and used to define a journey and reliability

ratio specific for each affected user. The first ratio defines the travel time change experienced by the users (Figure 4). The higher the journey ratio, the higher the users' relative increase in travelling time after the disruption. The second ratio assesses the difference of the users' path exposure to the hazard (Figure 5). When users access an infrastructure using the shortest path, they use road segments having a susceptibility to be affected by the hazard ($h_i$). The sum of the road segments' susceptibility for all segments taken by users to access the closest facility ($H_i$) is considered in this study as being

representative of the users' path reliability. It therefore represents the confidence that a user is not being impacted by the hazard when following the shortest path to access the closest facility (Immers et al., 2004; Jenelius et al., 2006). Comparison of the users' path exposure before and after the disruption enables to define a reliability ratio. The lower the ratio, the safer the path taken by the users relative to the original path.

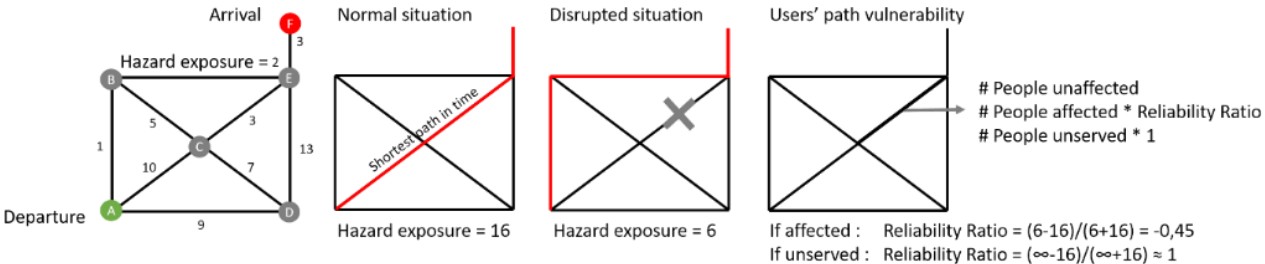

**Figure 5 - Accessibility assessment before and after a disruption using Dijkstra's shortest path algorithm. People affected and unserved are weighted by the Reliability Ratio which assesses how users' path exposure changes when taking the alternative shortest path in time to access the infrastructure.**

### 3.4. The metrics

#### 3.4.1. Road accessibility risk

The road accessibility risk ($Risk_i$) combines road access vulnerability ($V_i$) with the road segment's susceptibility to be affected by the hazard ($h_i$).

$$Risk_i = V_i * h_i ,$$ (2)

where $n$ represents the total number of villages in the study area.





The road access vulnerability index ($V_i$) summarizes the impact of a road segment's obstruction on the population's accessibility to the closest infrastructure (Equation 3). Population that will be affected by a rerouting ($\text{pop}_{affected}$) and population completely disconnected from access to services ($\text{pop}_{unserved}$) are both integrated into the index. The $\text{pop}_{j,affected}$ is weighed based on the *Journey Ratio$_j$* (Equation 4, see also Figure 4), which represents for each village ($j$) the journey difference in time (s) before ($t_{j,normal}$) and after ($t_{j,disturbed}$) the road segment obstruction. The journey ratio varies between 0 and 1 representing none to large changes in the journey, respectively. The $\text{pop}_{unserved}$ is weighed by one as this is the worse case scenario. The vulnerability index ($V_i$) varies from 0, for road segments with no impact on people's accessibility, to a maximum of 1. In this last situation, the entire population of the island would have no access to services through the obstruction of the segment, which in practice can only occur if service supply would be restricted to one location that is completely cut off from the road network.

$$V_i = \frac{\sum_{j=1}^{n}(\text{pop}_{j,affected} * \text{Journey Ratio}_j) + \text{pop}_{j,unserved}}{\text{pop}_{total}}, \tag{3}$$

$$\text{Journey Ratio}_j = \frac{t_{j,disturbed} - t_{j,normal}}{t_{j,disturbed}}. \tag{4}$$

Knowing the road access vulnerability ($V_i$) for each segment, it is then combined with to the road segment's susceptibility to be affected by the hazard ($h_i$) (see section 3.2) to obtain the road accessibility risk ($Risk_i$) caused by the obstruction of road segment *i* according to Eq. 2.

### 3.4.2. Users' path vulnerability

Similarly to the road access vulnerability index, the users' path vulnerability metric ($V_{u,i}$) integrates the population affected by a rerouting ($\text{pop}_{affected}$) and the population completely disconnected ($\text{pop}_{unserved}$). But instead of weighting the $\text{pop}_{affected}$ based on the journey travelling time difference, the affected population is weighed by the *Reliability Ratio$_j$* (Equation 5). Considering the user's path exposure as being the sum of susceptibility values for all segments crossed by a user during his journey to the closest service ($H_i$), this ratio records changes in the users' path exposure to the hazard before and after road segment obstruction (Equation 6). The *Reliability Ratio$_j$* varies between -1 and 1 as the alternative path, users are forced to take, may consist of road segments with an overall lower or higher exposure ($H_{j,disturbed}$) to the hazard compared to the original path ($H_{j,normal}$). If the alternative road improves the path's reliability, i.e. the user is forced to take a path with a lower chance of being impacted by the hazard, the ratio will be negative. The ratio will be equal to zero if the users' path reliability remains the unchanged before and after the segment interruption. In case the hazard susceptibility of the alternative path is higher than in the normal situation, the ratio is positive. Again, the $\text{pop}_{unserved}$ is weighed by one as this corresponds to the worst case scenario. The users' path vulnerability ($V_{u,i}$) similarly varies between -1 and 1.





$$V_{u,i} = \frac{\sum_{j=1}^{n}(pop_{j,affected} * \text{Reliability Ratio}_j) + pop_{j,unserved}}{pop_{total}},$$ (5)

$$\text{Reliability Ratio}_i = \frac{H_{j,disturbed} - H_{j,normal}}{H_{j,disturbed} + H_{j,normal}}.$$ (6)

### 3.5. Modelling assumptions

Accessibility assessment is a process that is computationally demanding (Postance et al., 2017). The number of alternative paths increases with the size of the road network, the number of villages and the number of infrastructures to process. While integrating travelers' demand (in this case access to the closest hospital) into the road segment analysis already leads to a more realistic representation of accessibility (Jenelius and Mattsson, 2015; Taylor and Susilawati, 2012), the following assumptions have been made in this study to make the computation feasible:

- The disruption affects only one segment at a time and is considered to last long enough. People will therefore not postpone their departure and will adapt to the situation by using alternative roads to access the closest infrastructure (Jenelius and Mattsson, 2015).
   - Time is the only element influencing their decision to access an infrastructure.
   - Users have a perfect knowledge of the network and know which is the shortest alternative if an alternative exists
(Jenelius and Mattsson, 2015).
   - All roads can be used in either direction.
   - The travel demand is constant at any time of the day and at any moment of the year, even during a disruption (Jenelius and Mattsson, 2015). At the time of the disruption, all inhabitants are residing in their home village. The hazard does not cause fatalities that will affect the travel demand.
- The disruption induces no congestion (Jenelius and Mattsson, 2015) which is a reasonable assumption for Ngazidja Island where the vehicle fleet is limited outside the capital.
   - The road network capacity is adapted to the travel demand.

## 4.   Results

### 4.1.  Road accessibility risk

#### 4.1.1. Road access vulnerability

In the context of Ngazidja Island, the road access vulnerability ($V_i$) analysis highlights that the most vulnerable roads are located in the close surrounding of the hospital facilities (Figure 6). Since close to facilities the number of travelers using these segments is important (Figure 6a), closure of a road segment close to the hospital will have a large impact on accessibility for a large part of the population. Road segments with limited road alternatives (Figure 6b) and the first segment of dead-end roads
(Figure 6c) are also characterized by higher road access vulnerability. When road redundancy is limited, the users' travelling time inevitably increases as it forces users to take a longer alternative route. Dead-end roads represent the only connection of



villages to the network. Their closure directly induces that the whole population of a village loses its road access to the hospital ($pop_{unserved}$). Idjikoundzi and Maouéni, for example, are two villages located high up on the volcano flank. Both villages depend on a single road segment to reach key infrastructure (Figure 6c). If this segment would become obstructed by a lava flow, it would prevent evacuation to hospitals or direct provision of emergency help to these villagers.

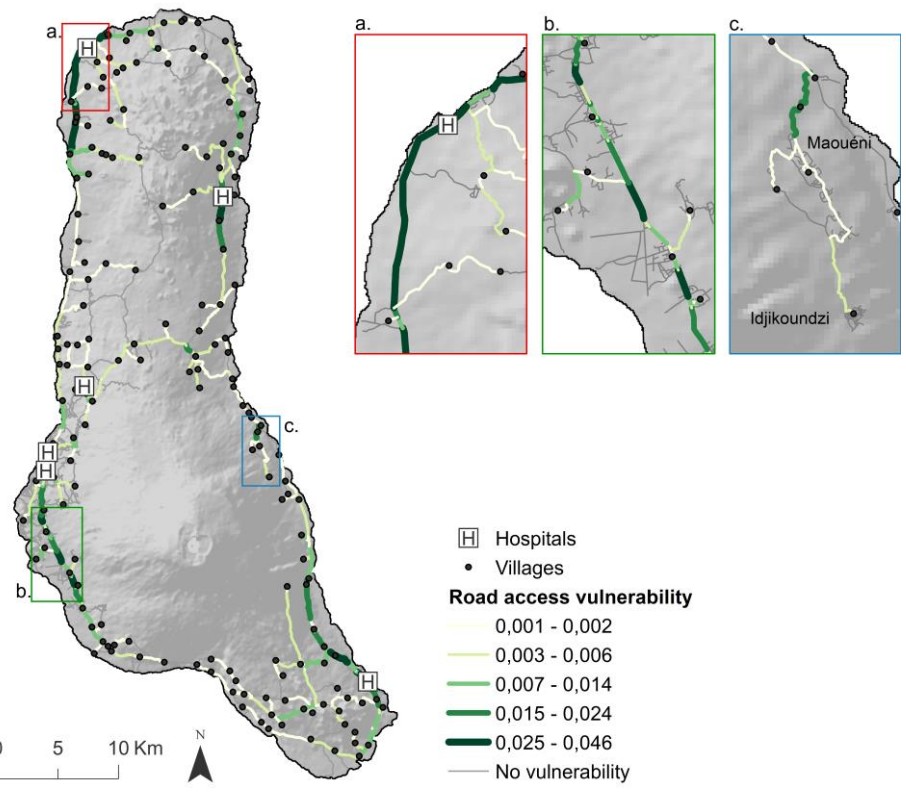

**Figure 6 - Road access vulnerability map representing the impact of each road segment obstruction on the population's accessibility to the closest infrastructure in terms of travel time.**

### 4.1.2. Road hazard exposure

10    Figure 7 shows the susceptibility of each road segment of being affected by lava flow ($h_i$) calculated as the sum of normalized hazard probability values occurring along the segment. The road segments characterized by the highest susceptibility are located on the northern flank of the Karthala massive and in the south. It concerns roads at higher elevation close to the volcanic rift zones along which most eruptions are initiated. Road segments located in the region of the 1977 eruption, near Singani and Hetsa villages, also present high chances of being affected by lava flow. Even if the probability for lava to outflow from "la

15    Porte d'Itsandra" and overflow the downstream areas is high (Figure 3), the probability decreases with distance. The probability that such flows reach road segments in the capital city Moroni is rather small. Finally, because the hazard exposure for each





segment, referred to as susceptibility here, is defined by summing up the normalized probability values of all pixels underlying the road segment, the exposure to hazard is strongly influenced by the segment length (Figure 7). Short segments as observed in the capital and in most villages (e.g. Koimbani - Figure 7) will have low susceptibility values, even if the area is highly exposed.

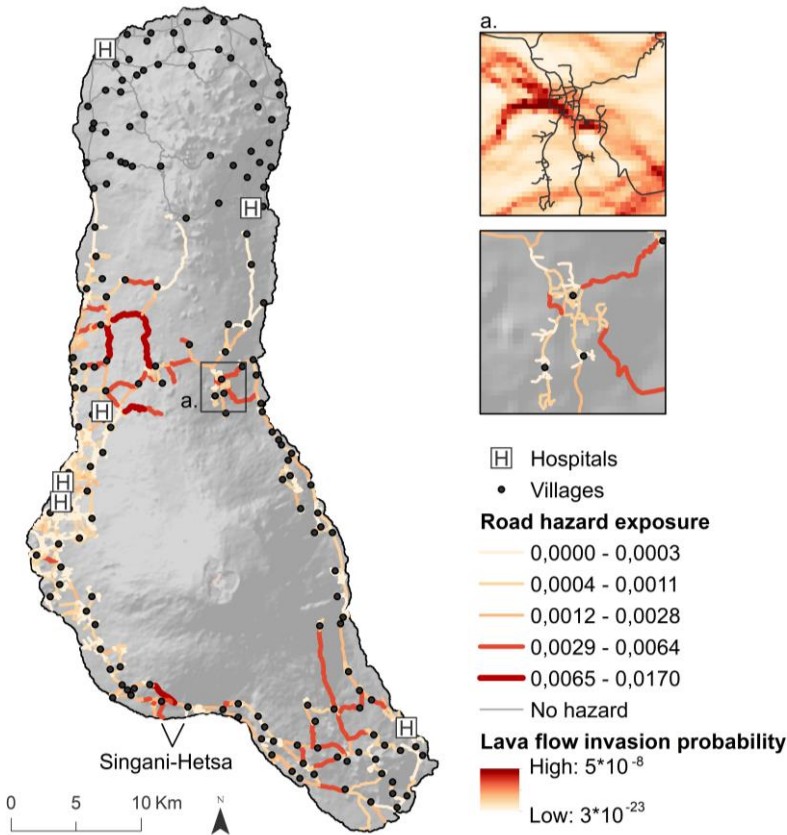

**Figure 7 - Road hazard exposure map showing susceptibility to lava flow hazard for each road segment with an inset on Koimbani. The upper inset represents the lava flow hazard probability map in the surrounding of Koimbani. Roads are overlaid on top of the hazard map. The lower inset shows susceptibility values for road segments in Koimbani.**

### 4.1.3. Road accessibility risk

10   Multiplying road access vulnerability for each segment ($V_i$) with the susceptibility of the road segment of being affected by a lava flow ($h_i$) enables identifying the accessibility risk associated with each road segment (Figure 8). Even if the northern part of the island is formed by La Grille volcano (Figure 1), roads at risk are concentrated around Karthala volcano which is the volcano for which lava flow hazard is the highest. Road segments with a high accessibility risk are segments with a limited number of alternative roads (road redundancy) in the immediate surroundings (Figure 8a), first segments of dead-end roads

15  (Figure 8b), and roads associated with a high lava flow susceptibility (Figure 8c).



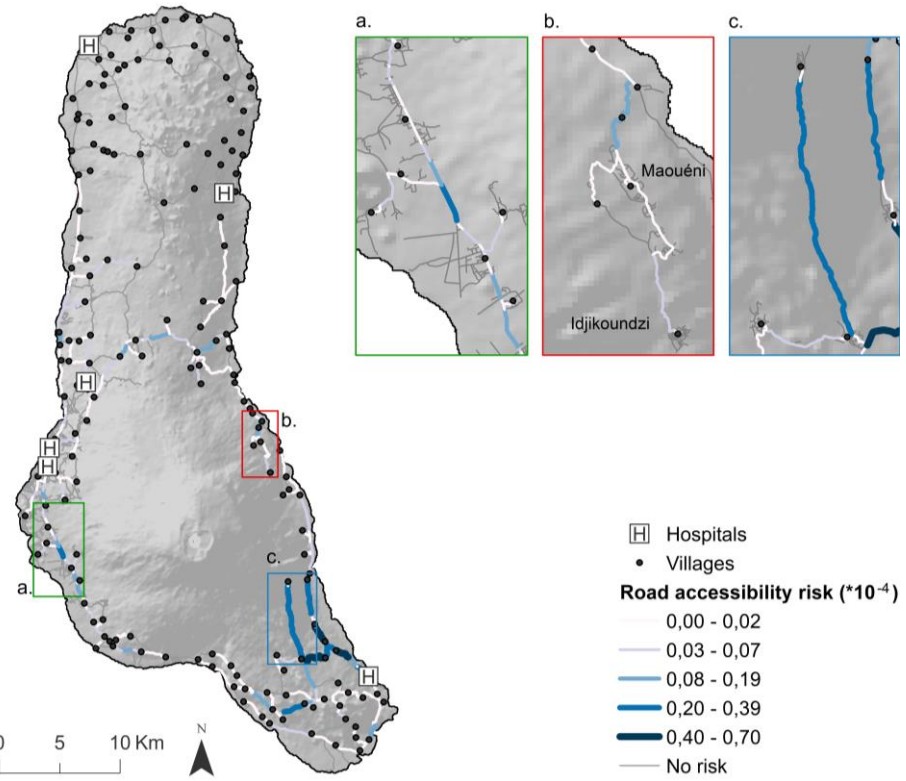

**Figure 8 - Accessibility risk map for road segments combining road access vulnerability based on travel time (s) to the closest infrastructure with road segment susceptibility to lava flows.**

### 4.2. Users' path vulnerability

5   Integrating users' exposure to hazard while travelling to the closest facility enables to identify whether the alternative shortest path in case of a road segment obstruction increases or decreases hazard exposure. The former results in an increase in users' path vulnerability, the latter in a decrease. Main road segments located close to and within the area potentially affected by Karthala lava flows show high users' path vulnerability index values ($V_{u,i}$) (Figure 9a-b-c) since closure of these road segments would induce some users to choose for a less reliable route Moreover, it is also interesting to note that the users' path

10  vulnerability of some road segments is negative (Figure 9d). This indicates that the alternative shortest path taken by most users that would normally be traversing this road segment is associated with a lower probability to be interrupted by lava flow, and that the reliability of the route for these users therefore improves.





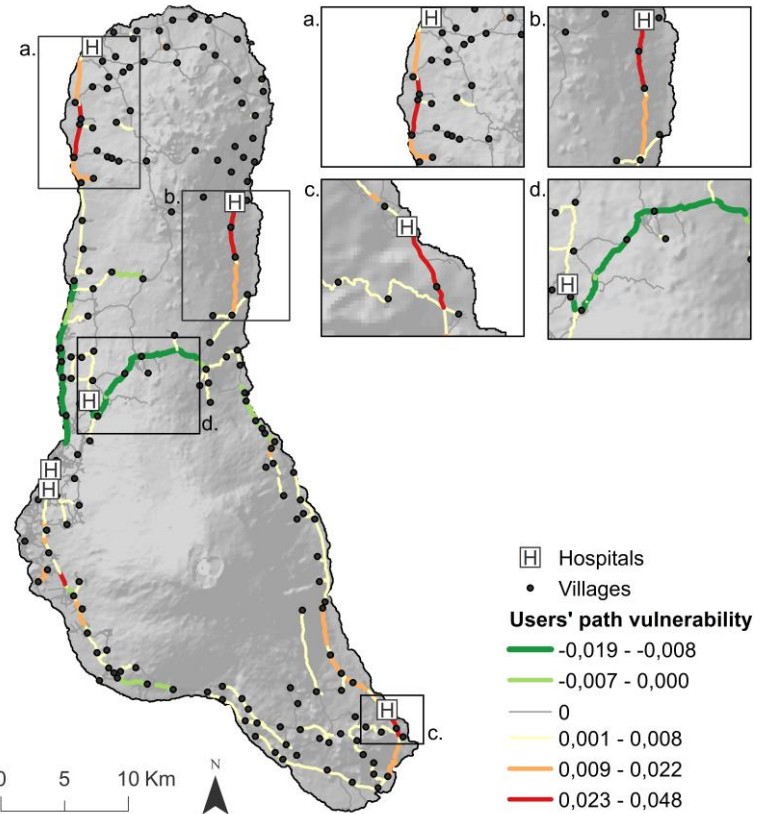

**Figure 9 - Road users' path vulnerability.**

## 5. Discussion

The metrics proposed in this study provide a good overview of which road segments are the most strategic in a road network
in terms of impact of road closure on access to key infrastructure in case a hazard occurs.

For long term management, the road accessibility risk metric (1) identifies road segments that are exposed to hazard and that
would substantially reduce access to infrastructure if being obstructed by a lava flow and (2) enables to quickly calculate and
visualize the impact of any changes proposed in the road network (e.g. in the network's structure or the segment's
characteristics) on accessibility and exposure. Such impacts must be carefully interpreted and discussed since modifications
of the road network have a cost and even if the risk in some parts of the region may be reduced, it can be at the expense of a
higher risk in other locations.

To lower the risk associated with each road segment, measures can be taken to prevent or reduce the impact of specific hazards
(e.g. rock safety net, drainage channel, dykes). Engineering works can also make the roads more hazard-proof or reduce the
hazard's direct physical impact. In the case of lava flows, however, little can be done to prevent the hazard or to adapt road





infrastructure to better cope with the hazard. Measures should therefore focus on reducing the impact of road obstruction on the population's accessibility. Modifying the population demand on the network can, for example, reduce road access vulnerability associated with particular road segments. For the current situation, the overall road access vulnerability index associated with the road network within the service area of the Foumbouni hospital, for example, is equal to 4.99 (Figure 10a).

As it can be observed in Figure 10b, the construction of a new road segment has a positive impact on the overall road access vulnerability associated with the road network in the service area. People which were before isolated or highly impacted due to a road obstruction now have an alternative path to the hospital. Redundancy positively influences road access vulnerability (Mattsson and Jenelius, 2015; Taylor et al., 2006) and lowers the overall road access vulnerability index to 4.45. The development of new infrastructure can also positively influence road access vulnerability (Figure 10c). A new hospital location

within the service area of Foumbouni hospital reduces overall road access vulnerability within the area to 4.09.

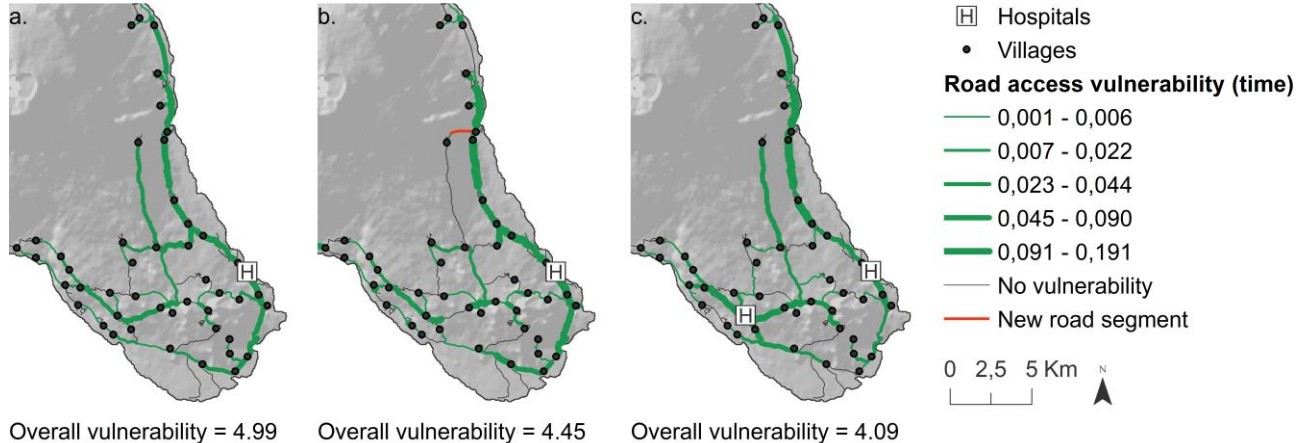

**Figure 10 - Modification of the actual road access vulnerability (a) by constructing a new road segment (b) or by developing a new infrastructure (c). The overall road access vulnerability is the sum of the road access vulnerability values of all road segments within**
**the Foumbouni hospital service area.**

Improvement of the speed can finally also contribute to changes in road access vulnerability. Suppose that the average speed of 27 ± 10 km/h reached on national roads is improved to 70 km/h through increased road width, better road conditions and maintenance. The proposed methodology then allows a quick evaluation of the impact on road access vulnerability. While increasing the speed does not allow reducing the overall road access vulnerability it tends to concentrate traffic flow on the

fastest road segments (Figure 11) (Taylor et al., 2006). Alternatively, road segments least exposed to hazard could be preferentially improved to increase the effective speed on these roads. This would re-direct users to these more rapid roads and decrease the risk of users being impacted by a road obstruction due to a hazard.



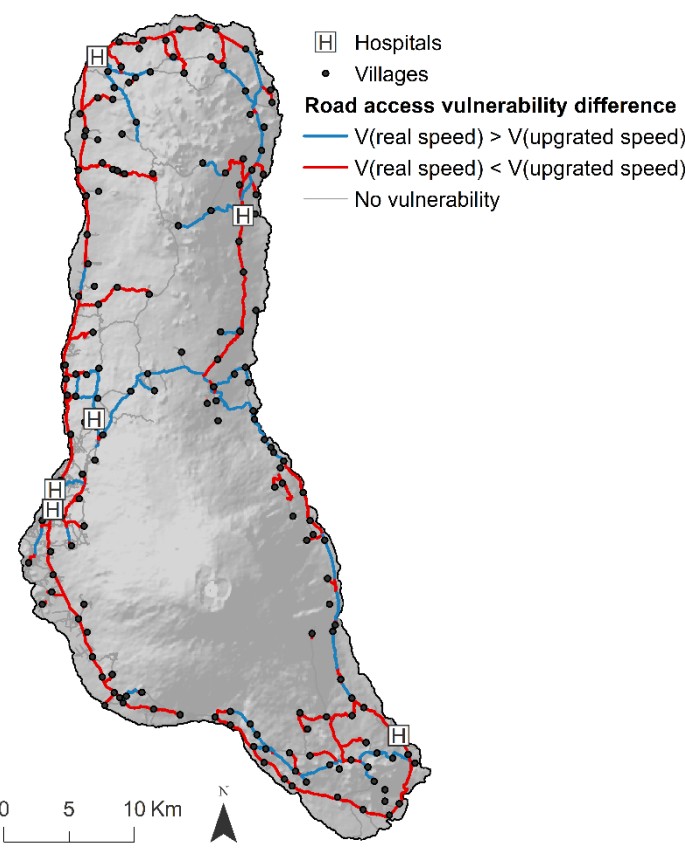

**Figure 11 - Impact on road access vulnerability of an improvement of the speed that can be reached on national roads.**

The second metric presented in this research, the users' path vulnerability metric, defines the impact the closure of a road segment would have on the reliability of the alternative route in terms of exposure to hazard. This metric can be used for short term management as it identifies the road segments increasing or lowering the exposure of users to the hazard when travelling to the closest infrastructure. It can be used as an argument to:

- Keep some road segments open: close to the border of the area that is potentially affected by Karthala lava flows, road segments have high users' path vulnerability index values (Figure 9a-b). Closure of these road segments would induce some users to have to choose for a less reliable route. Indeed, if these road segments become obstructed, people will be forced to take an alternative route, principally more to the south, which is more exposed to lava flow hazard (Figure 3).
- Prioritize repair of obstructed road segments: road segments within the area that is potentially affected by Karthala lava flows and that are associated with a high users' path vulnerability index (Figure 9c) may be prioritized for repair if obstructed at the same time as other segments, if the modeling shows that their obstruction would force people to expose themselves to a higher hazard probability when taking the detour route.




- Preventively close some road sections: road segments with a negative users' path vulnerability may be closed in case an event is expected in the service area containing the road segment, in order to re-orientate the flow of users to less exposed roads (Figure 9d).

## 6. Perspectives

Modelling of accessibility in case of a hazard, as proposed in this study, assumes that the disruption of the road network affects only one segment at a time. Future improvements of the model should be able to deal with obstruction of multiple segments, as lava flows can affect different road segments while following their path or develop branches in different directions. Attention should also be given to a proper representation of the road network and its attributes, as in other potential study areas road networks can be more complex than on Ngazidja Island (e.g. one direction roads, turn restrictions, overpasses, tunnels…).

For the sake of illustration, accessibility to only one type of service (i.e. hospital) was discussed, without taking into account functional characteristics or capacity constraints. It would be interesting to generalize the proposed methodology to be able to concurrently cope with different types of services (e.g. emergency services, economic activity…) and to integrate the capacity and attractiveness of specific infrastructures using a gravity modelling approach (Guagliardo, 2004; Luo and Qi, 2009). As such, other characteristics governing the choice of infrastructure can be incorporated in the analysis.

One should also keep in mind that in our analysis travel demand is assumed constant at any time of the day, week, and at any moment of the year, and during a disruption, and that no congestion effects are taken in account. Taking diurnal, weekly and seasonal differences in travel demand and movement patterns in account would require detailed information about human activity patterns and travel behavior, which was not available for this study. However, with the rise of big data on resident's mobility patterns one may expect that more realistic scenario analysis of impacts of road obstruction on accessibility will

become feasible, also in the case of a hazard. Implementation of these modelling perspectives would contribute to a more realistic simulation of road segments' importance and users' behavior on the road network.

Finally, the analysis presented in this research focuses on lava flow hazard only. Yet other hazards are capable of blocking a road as well (e.g. ash fall, floods, rock fall…). Lava flows are natural hazards that have a high destructive power on infrastructure. When a lava flow hits a road, it is realistic to assume that the road is completely obstructed. But this is not

always the case with other natural hazards. In these situations, when a road is affected, users might be able to still use the road segment. Based on the type and intensity of the hazard, the users' travel time will be affected to some extent. Speed disruption functions or functional losses described in other studies (Jenkins et al., 2015; Pregnolato et al., 2017; Wilson et al., 2012) might be integrated in the shortest path analysis and therefore enable to adapt the proposed methodology to a wider range of hazards (e.g. ash fall, congestion…).



## 7. Conclusions

Accessibility analysis in a hazardous situation is important since the population must be able to effectively evacuate, access shelters or medical infrastructure, while emergency services must be able to assist the population in situ. For hazards with a spatially heterogeneous probability of occurrence, combining hazard maps with road network accessibility measures provides

a new way to support functional risk assessment.

The current study presents two metrics to assess the importance of a road segment on people's mobility effectively using location specific information provided by probabilistic hazard maps. Both metrics enable quantitative assessment of impacts of hazard on accessibility to critical infrastructure and result in maps that may support and feed discussions about the development of new infrastructure, road capacity increase, maintenance and emergency procedures.

The road accessibility risk metric proposed in this study identifies the impact of each road segment's obstruction on the population's accessibility to infrastructure and combines it with the road segment's susceptibility of being affected by a hazard. The users' path vulnerability metric defines the impact the closure of a road segment has on the reliability of the alternative route a user of the road network would be forced to take. This metric highlights that not only roads in a hazardous environment must be considered in risk assessment, also road segments located further away from the hazard may be important as their

obstruction may force people to take an alternative road which is less reliable. User's path vulnerability may also be used to prioritize the re-opening of affected segments or to preventively close some road segments located in the proximity of an expected hazard to re-orientate users to less exposed roads. The two metrics presented in this study may contribute to a rational analysis of accessibility related risks of natural hazards and scenario analysis for reducing these risks.





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
