# Peer review of "Assessing the impact of road segment obstruction on accessibility to critical services in case of a hazard"

_Natural Hazards and Earth System Sciences, 2018_

## Referee Comment (RC1) · Anonymous Referee #1 · 10 Feb 2019

The paper analyses the impact of a hazard on the road network using two metrics, (i) road accessibility risk and (ii) road hazard exposure. The first combines the road segment's potential impact on the accessibility and travelling time of the population to the closest infrastructure ("road access vulnerability") with the probability of occurrence of the hazard ("road hazard exposure"). The second metric considers the reliability of the alternative path the user needs to follow in case a road segment becomes disrupted. The content of the manuscript fit the scope of the journal. The structure is good and the argument analysed is interesting and novel. Despite some limits of the analysis due to many modelling assumptions, anyway well discussed and argued by the authors themselves, in my opinion the paper can be accepted to the publication on NHESS with some minor integration. The greatest doubt is in the way in which the Road Hazard

[Figure]

Exposure is estimated. In the chapter Road Hazard Exposure authors state: "referred to as susceptibility here, is defined by summing up the normalized probability values of all pixels underlying the road segment, the exposure to hazard is strongly influenced by the segment length (Figure 7). Short segments as observed in the capital and in most villages (e.g. Koimbani - Figure 7) will have low susceptibility values, even if the area is highly exposed". Instead of the sum of the probability value on the road segment, others metrics to evaluate the susceptibility could be more appropriate (ie. assigning the modal value of the probability to the segment or an x percentile of the probability) producing a result without anomalies related to the high influence of the segment length. I would like authors reply to this remark also in the paper, or better, they could try to use an other metric to estimate the road hazard exposure also in the analysis. An other issue is related to the value assigned to the road in the northern part of the island. Since the analysis is restricted to the lava flow produced by the Karthala volcano, in the roads of the northern part the value should be "not applicable" and not "no hazard". I do not think it is acceptable assign the value "no hazard" since in the northern portion of the island there is another volcano, which could subject the streets to lava flows. In figure 3 it seems that the roads and inhabitants in the northern part of the island would not be affected by the lava flows of the Karthala volcano.
* * *

---

## Referee Comment (RC2) · Anonymous Referee #2 · 18 Feb 2019

The authors of this paper have had a good inspiration focusing their analysis on the road networks impact on accessibility of the people at the emergency services in a case of hazard. Nevertheless, there is a rich literature on the prevention and mitigation of the impact of different hazards, there are still many things to be done for a useful mapping of the relationships between three pillars: location of the emergency services, hazard probability and the impact of the roads on rapid accessibility of population to medical assistance. The paper is very well-written, with a synthetic abstract for the reader to be interested in the content. After crossing the relevant literature in the field, there is the necessary framework to insert own results in the international ideas' circuit which ensures the guaranty of a general scientific interest. There are some strong points of this paper: a) the methodology, based on two metrics (road accessibility risk

and users' path vulnerability), which is clear and applicable in a diversity of hazards; b) clear results, reflected by specific maps; c) utility of the conclusions for the next academic development and for a better functional risk management. This paper offers practical ideas to rethink and improve the entire road network of Ngazidja Island. The figure 10 shows the necessity to multiply the medical services' locations in some strategic points, and to reevaluate the speed on the roads for a rapid accessibility to emergency services. Through the methodology, the results and especially the practical importance of such studies, the paper deserves to be published in the journal. So, I highly recommend it for publication with minor suggestions. First, I believe that the title needs to be clearer. I understand that the authors wish to reveal the importance of the analysis of the road networks at the segment level for a strategic risk management in the case of a hazard. But why is necessary to analyze an entire network, when is possible to individualize the possible affected segments? So, my proposal is to modify the title which could have the next form: Assessing "the impact of affected road segment" on accessibility. . . . . . Secondly, please specify clearer what means road segment? In your paper, you mention individual road segment! It's about the road between the successive villages? Analyzing the maps, it results that the road segment is defined by two nearest road connected villages on the same road. Usually the road segment is a very relative term, connecting two villages at variable distance, or between two different localities. Thirdly, the authors should appreciate better the road access vulnerability impact in a case of hazard and to have a general image on a possible disaster, technically speaking, there is the possibility to increase the maps relevance, introducing the population size of the localities.
* * *

---

## Author Comment (AC2) · 18 Apr 2019

Comment 1: First, I believe that the title needs to be clearer. I understand that the authors wish to reveal the importance of the analysis of the road networks at the segment level for a strategic risk management in the case of a hazard. But why is necessary to analyze an entire network, when is possible to individualize the possible affected segments? So, my proposal is to modify the title which could have the next form: Assessing "the impact of affected road segment" on accessibility......

Response: Thank you for this suggestion. We agree that the title could be made clearer.

Changes in manuscript: We propose to change the title as follows: "Assessing the

[Figure]

impact of road segment obstruction on accessibility to critical services in case of a hazard".

Comment 2: Secondly, please specify clearer what means road segment? In your paper, you mention individual road segment! It's about the road between the successive villages? Analyzing the maps, it results that the road segment is defined by two nearest road connected villages on the same road. Usually the road segment is a very relative term, connecting two villages at variable distance, or between two different localities.

Response: In our study road segments are defined as the edges between each pair of nodes in the road network. A node corresponds to a location in the road network where two roads cross or come together or to the location of a village or a health service from/to which accessibility is assessed. This is explained in section 3.2, lines 10-13.

Comment 3: Thirdly, the authors should appreciate better the road access vulnerability impact in a case of hazard and to have a general image on a possible disaster, technically speaking, there is the possibility to increase the maps relevance, introducing the population size of the localities.

Response: We are not quite sure what is meant by this remark/suggestion. If the remark refers to the fact that we did not include information on the population size of the villages in the map of accessibility risk (figure 7), we would like to make clear that we did not do so to not overload the maps with information, given the large number of villages. After all, population size of the villages affected is taken in account in the calculation of both indicators. For information on population size of the villages the reader is referred to figure 2. If the remark does not refer to the way accessibility risk has been mapped, then please let us know and provide us more info on the exact meaning of the remark.

---

## Author Response (AR1)

**Paper nhess-2018-379**

**Response to reviewer comments**

Reviewer 1 (RC1)

1. *The greatest doubt is in the way in which the Road Hazard Exposure is estimated. In the chapter Road Hazard Exposure authors state: "referred to as susceptibility here, is defined by summing up the normalized probability values of all pixels underlying the road segment, the exposure to hazard is strongly influenced by the segment length (Figure 7). Short segments as observed in the capital and in most villages (e.g. Koimbani - Figure 7) will have low susceptibility values, even if the area is highly exposed". Instead of the sum of the probability value on the road segment, other metrics to evaluate the susceptibility could be more appropriate (ie. Assigning the modal value of the probability to the segment or an x percentile of the probability) producing a result without anomalies related to the high influence of the segment length. I would like authors reply to this remark also in the paper, or better, they could try to use another metric to estimate the road hazard exposure also in the analysis.*

While doing our research we carefully considered how to include road hazard exposure in the modelling. We looked at the possibility of using the mean or median probability of the pixels constituting each road segment, as well as using a percentile of the probability values as a way of quantifying road hazard exposure. In all these cases however no consideration is given to the fact that if a lava flow occurs at one specific location along the segment, the entire road segment will be blocked, and that if two locations along a segment have a high probability of being affected by a lava flow, chances that the segment will effectively be blocked by lava flow will therefore be substantially higher than if high probabilities only occur at one location along the segment. As such longer segments indeed have a larger chance of being affected.

We have modified the text in section 4.1.2 as follows: "Because the hazard exposure for each segment, referred to as susceptibility here, is defined by summing up the normalized probability values of all pixels underlying the road segment, exposure of a segment to hazard is influenced both by probability values at pixel level as well as by the length of the segment. This reflects the fact that longer segments having a chance of being affected by lava flows at different locations along the segment are also more exposed to lava flow hazard and thus have a higher chance of being blocked by one or more lava flows. Accordingly, shorter road segments as observed in the capital and in most villages (e.g. Koimbani – Figure 7) will have lower susceptibility values than some longer segments, even if the area is highly exposed."

2. *Another issue is related to the value assigned to the road in the northern part of the island. Since the analysis is restricted to the lava flow produced by the Karthala volcano, in the roads of the northern part the value should be "not applicable" and not "no hazard". I do not think it is acceptable assign the value "no hazard" since in the northern portion of the island there is another volcano, which could subject the streets to lava flows. In figure 3 it seems that the roads and inhabitants in the northern part of the island would not be affected by the lava flows of the Karthala volcano.*

The zero value for hazard exposure for the road segments shown in figure 7 specifically refers to lava flow hazard of the Karthala volcano. We only considered the Karthala in our study since the volcano situated in the northern part of the island (La Grille) is not considered as an active volcano: there is no clear evidence of eruptions during the Holocene (Global Volcanic Program: https://volcano.si.edu/volcano.cfm?vn=233001). Since La Griffe is not an active volcano and the road segments in the northern part of the island cannot be reached by lava flows emerging from the Karthala we do not think the use of the nil value in figure 7 is misleading.

In order not to create confusion we have clearly indicated in the figure captions of figures 7-8 that the values mentioned on the map refer to lava flow hazard exposure of the Karthala:

"Figure 7 – Road hazard exposure map for the Karthala volcano showing susceptibility to lava flow hazard for each road segment with an inset on Koimbani."

"Figure 8 – Accessibility risk map for road segments combining road access vulnerability based on travel time (s) to the closest infrastructure with road segment susceptibility to Karthala lava flows."

Reviewer 2 (RC2)

1. *First, I believe that the title needs to be clearer. I understand that the authors wish to reveal the importance of the analysis of the road networks at the segment level for a strategic risk management in the case of a hazard. But why is necessary to analyze an entire network, when is possible to individualize the possible affected segments? So, my proposal is to modify the title which could have the next form: Assessing "the impact of affected road segment" on accessibility......*

Thank you for this suggestion. We agree that the title could be made clearer.

We have changed the title as follows: "Assessing the impact of road segment obstruction on accessibility to critical services in case of a hazard".

2. *Secondly, please specify clearer what means road segment? In your paper, you mention individual road segment! It's about the road between the successive villages? Analyzing the maps, it results that the road segment is defined by two nearest road connected villages on the same road. Usually the road segment is a very relative term, connecting two villages at variable distance, or between two different localities.*

In our study road segments are defined as the edges between each pair of nodes in the road network. A node corresponds to a location in the road network where two roads cross or come together or to the location of a village or a health service from/to which accessibility is assessed. This is explained in section 3.2, lines 10-13.

3. *Thirdly, the authors should appreciate better the road access vulnerability impact in a case of hazard and to have a general image on a possible disaster, technically speaking, there is the possibility to increase the maps relevance, introducing the population size of the localities.*

We are not quite sure what is meant by this remark/suggestion. If the remark refers to the fact that we did not include information on the population size of the villages in the hazard/risk/vulnerability maps (figures 7-9), we would like to make clear that we did not do so to not overload the maps with information, given the large number of villages. Also, population size of the villages affected is explicitly taken in account in the calculation of the two indicators proposed in this study, so the effect of population size is included in the risk and vulnerability maps shown (figures 8-9).

**Response to editor comments**

*Both reviewers suggested minor revisions. The editor considers that the response in the interactive discussion is proper to reviewer 1. The changes should be implemented as suggested. In case of reviewer 2 there is an unclarity regarding the 3rd comment. The editor considers that maybe population size could be reflected by the size of dots among different thresholds, if possible with the employed software. If this is not possible, the responses seem acceptable but this should be listed among the assumptions and the other suggested changes (title) should be implemented. The explanation to the 2nd comment is acceptable. The editor has nothing significant to add to the comments of the referees and agrees with the comments done.*

All changes listed in our response to the comments made by the reviewers (see above) have been implemented in the new version of the manuscript.

Regarding the inclusion of information on population size of the villages in the road hazard exposure, road accessibility risk and road user's path vulnerability maps (Figures 7-9), we feel that including this information in the maps leads to overloading, obscuring the information at road segment level, as mentioned in our response to the reviewer (see above). We have done some experiments by adding proportional symbols showing population size of the villages on Figure 7 (see below). We experimented with different circle sizes and line widths for the outline of the circles, yet we were never quite satisfied with the result and would therefore prefer to stick to the figures as shown in the original manuscript. If the reader would like to have information on village size, we suggest use of figure 2 in combination with figures 7-9.

[Figure]

[revised manuscript text omitted]
, exposure of a segment to hazard is influenced both by probability values at pixel level as well as by the length of the segment. This reflects the fact that longer segments having a chance of being affected by lava flows at different locations along the segment are also more exposed to lava flow hazard and thus have a higher chance of being blocked by one or more lava flows. the exposure to hazard is strongly influenced by the segment length (Figure 7). Accordingly, Short shorter segments as observed in the capital and in most villages (e.g. Koimbani - Figure 7) will have lower susceptibility values than some longer segments, even if the area is highly exposed.

[revised manuscript text omitted]